# Effectiveness of a pharmacist diabetes coaching program: A propensity-matched retrospective analysis

Daniel Amante[1]*, Samir Malkani[2,3], Richard Haas[2,3], Biqi Wang[4], Cheryl Barry[3], Bill McElnea[5], Lillian Piz[5], Gabriella Pugliese[5], Hinal Sharma[5], Apurv Soni[4]

1 Department of Population and Quantitative Health Sciences, Division of Health Informatics and Implementation Science, UMass Chan Medical School, Worcester, Massachusetts, United States of America, 2 Department of Medicine, Division of Endocrinology & Diabetes, UMass Chan Medical School, Worcester, Massachusetts, United States of America, 3 Diabetes, Endocrinology, and Metabolism, UMass Memorial Health, Worcester, Massachusetts, United States of America, 4 Department of Medicine, Division of Health System Science, UMass Chan Medical School, Worcester, Massachusetts, United States of America, 5 Shields Health Solutions, Stoughton, Massachusetts, United States of America

* Daniel.amante@umassmed.edu

## Abstract

### Background

Limited evidence exists on the clinical and cost impact of diabetes coaching programs integrated into care teams. The Diabetes Care Coach (DCC) program is a pharmacist-driven telehealth coaching program for patients with poorly controlled diabetes. Coaches provide frequent telehealth support to improve glycemic control through medication management, nutrition and lifestyle counseling, leveraging diabetes technologies, initiating mental health referrals, and addressing health-related social needs. The objective of this study was to evaluate the effectiveness of the UMass Memorial Health (UMMH) DCC program.

### Methods and findings

A retrospective matched cohort study was conducted with data collected from the UMMH electronic health record and the UMass Memorial Medicare Accountable Care Organization (ACO) claims database from January 2020 to December 2023. The DCC program was implemented at the UMMH diabetes clinic, a specialty care clinic in Worcester, MA, supported by the UMass Specialty Pharmacy program, which is managed by Shields Health Solutions. Participants included patients with persistent severe hyperglycemia (hemoglobin A1c [A1c] ≥ 9) receiving care at the diabetes clinic. Intervention participants (n = 239) enrolled in the DCC program were matched with comparison participants (n = 815) not enrolled in the program during the study period. Intervention and comparison participants who were also enrolled in ACO were considered for cost analyses. Between-group differences in glycemic control (A1c

**Data availability statement:** Data cannot be shared publicly because they contain potentially identifying patient information derived from electronic health records and administrative claims and are subject to ethical and legal restrictions imposed by the UMass Chan Institutional Review Board and institutional data use agreements with UMass Memorial Health and the UMass Memorial Medicare Accountable Care Organization. Data are available from the UMass Chan Institutional Review Board (irb@umassmed.edu) for researchers who meet the criteria for access to confidential data.

**Funding:** The Diabetes Care Coach program is funded in part by Shields Health Solutions, a vendor to UMass Memorial Health. Shields-affiliated authors on this study have no other financial associations with UMass Memorial Health. This project was partially supported by the National Institute of Diabetes and Digestive and Kidney Diseases (NIDDK) Grant K01DK131318 (support for DA). AS has consulted for ARPA-H, Pfizer, FDA, and Shields Health Solutions on matters unrelated to this manuscript. This project was also partially supported by the National Center for Advancing Translational Sciences (NCATS), National Institutes of Health, through Grant UL1TR001453 (support for AS). The content is solely the responsibility of the authors and does not necessarily represent the official views of the NIH.

**Competing interests:** The authors have declared that no competing interests exist.

change), health care utilization (emergency department visits and days hospitalized), and cost (total medical expenditures, TME) were evaluated. Intervention participants experienced a greater mean A1c reduction of –0.4 percentage points (95% CI, –1.1 to 0.3) compared with comparison participants. Within the ACO subgroup, intervention participants also had a greater mean reduction in annual TME of $2,649 per patient (95% CI, –$14,425 to $9,127) and a reduction in hospital days per patient per year of –5.2 (95% CI, –9.8 to –0.6) relative to comparison participants.

## Conclusions

The DCC program was associated with directional improvements in clinical, health care utilization, and cost outcomes and was cost-neutral due to savings and associated revenue. With approximately one-third of participants having Medicaid insurance, the sustainability of such programs increases through leveraging the 340B program to enhance care for economically vulnerable patients.

## Introduction

Diabetes remains a formidable public-health challenge in the United States. Over 30 million adults live with diabetes, and nearly one in five (18%) have a hemoglobin A1c (A1c) level above 9%, a threshold that signals markedly poor glycemic control [1]. Individuals in this subgroup experience higher rates of hospitalization, longer inpatient stays, and greater medical expenditure than those whose diabetes is controlled [2]. Because uncontrolled hyperglycemia drives complications and cost, the U.S. Department of Health and Human Services has made "reducing the proportion of adults with A1c >9%" a core objective of its Healthy People 2030 campaign [3]. Yet national surveillance shows little to no progress toward that goal [4]. Persistent barriers, medication affordability and non-adherence, limited health literacy, lack of patient engagement, and social determinants of health such as food insecurity and transportation, continue to limit the impact of conventional clinic-based care [5].

To address these multidimensional needs, diabetes coaching models have demonstrated potential effectiveness. Randomized trials and meta-analyses have found that coaching, often delivered by pharmacists, nurses, or certified diabetes educators, can improve medication adherence, self-efficacy, and A1c while reducing diabetes-related distress [6–10]. Yet most of this evidence arises from within tightly controlled research settings with dedicated resources, protocolized encounter schedules, and selective patient samples. Whether similar benefits are achieved when coaching is embedded in clinical practice, financed through existing operational budgets, and delivered to a diverse sample of high-risk patients is largely unknown. Data are particularly scarce on the impact of coaching programs on downstream outcomes that matter to health-system leaders, such as changes in emergency department visits, hospital utilization, and total medical expenditures [11].

Despite the demonstrated benefits of diabetes health coaching interventions, several barriers hinder their long-term sustainability within healthcare systems. One

major challenge is securing consistent funding, especially when coaching programs are not directly reimbursed by insurance or tied to traditional fee-for-service models. Additionally, integrating coaches into existing care teams can be difficult due to unclear role definitions, limited clinical workflows that support team-based care, and variability in training and credentialing of coaches [12]. There may also be resistance from providers or administrators who are unfamiliar with the value of coaching or concerned about added workload and costs. Furthermore, scalability is often limited by inconsistent data systems that make it difficult to track outcomes and demonstrate return on investment. These barriers underscore the importance of strategic planning, leadership support, and policy changes to embed coaching more fully into sustainable, value-based care models.

Although coaching interventions have been associated with clinical improvements, there is a lack of research on the associated improvements in the cost of care [10].

For programs that are integrated into clinical care, it is important to identify factors associated with greater improvement and to understand the impact that such programs have on cost of care. In this paper, we:

1. Evaluate a diabetes coaching program that integrated clinical pharmacists into the care team at an academic-affiliated specialty diabetes clinic.

2. Compare clinical and cost outcomes with a propensity-matched cohort.

## Methods

### Study design and data sources

We performed a retrospective matched-cohort study comparing outcomes between patients enrolled in a diabetes coaching program and a comparison group of usual care patients. Data were obtained from the UMass Memorial Health (UMMH) electronic health record (EHR) for clinical outcomes and from the UMass Memorial Medicare Accountable Care Organization (ACO) claims database for cost outcomes. The study period spanned January 2020 through December 2023, capturing 1-year pre-intervention and 1-year post-intervention data for each participant. The data were accessed on April 25, 2025. The UMass Chan IRB reviewed the study (STUDY00000536) and granted both a HIPAA waiver and a waiver of informed consent. Reporting follows the STROBE guidelines for observational studies.

Leveraging a retrospective matched-cohort design, we compared adults with persistent A1c ≥ 9% who enrolled in the diabetes coaching program with propensity-matched peers who were also receiving specialty diabetes care but were not enrolled in the diabetes coaching program. In this manuscript, we evaluated whether diabetes coaching program participation was associated with greater improvements in A1c and changes in emergency department use, hospitalizations, and annual total medical expenditure (TME). We also explored whether effects differed for ACO patients versus non-ACO patients. By examining both clinical and economic end points, this manuscript seeks to provide real-world evidence that health system decision makers need to judge the value and sustainability of diabetes coaching.

### Setting

This study took place at the UMMH diabetes clinic which is a part of the UMass Memorial Diabetes Center of Excellence in Worcester, Massachusetts. Our diabetes clinic provides specialty and co-management care to over 10,000 individuals who reside primarily in Central Massachusetts. Specialty diabetes care is generally delivered by multidisciplinary teams consisting of endocrinologists, nurse practitioners, and diabetes care and education specialists.

The Diabetes Care Coach (DCC) program is a pharmacist-driven initiative at the UMMH diabetes clinic. Pharmacists were selected as diabetes coaches based on their specialized training in diabetes pharmacotherapy, medication titration, and medication adherence support, as well as their established role within specialty pharmacy and medication management workflows. The program was staffed by two pharmacist coaches providing approximately 2.0 full-time equivalent (FTE) effort dedicated to the DCC program during the study period. The coaches completed training on delivering the

DCC program over the course of 1–2 months as part of program onboarding. This training focused on enhancing clinical knowledge regarding medication and disease state management, shadowing of certified diabetes education and care specialists, providers, and pharmacy liaisons in clinic, learning specialty pharmacy dispensing system and workflow, and UMMH EHR system training. The coaches received additional training in diabetes self-management education and support (DSMES) as part of their ongoing clinical development and subsequently obtained Certified Diabetes Care and Education Specialist (CDCES) credentials. This training was not a formal component of the DCC intervention itself but reflected professional growth aligned with their clinical roles. Accordingly, DSMES training costs were not included in the financial analysis, as they were not program-specific implementation expenses.

The coaches (n = 2) provided weekly telehealth support for the first 1–2 months, then every other week for the remaining months of study participation, with frequency adjusted based on patient clinical needs. The telephone-based coaching appointments were up to 1 hour long for the initial visit and up to 30 minutes long for follow-up visits. The structure of the coaching visits included review and discussion of CGM or glucometer data, medication adherence, lifestyle and nutrition, and setting diabetes self-management behavioral goals. The coaching visits were intentionally flexible and tailored to patient clinical needs rather than delivered using a fixed protocol. The patient-to-coach ratio during the study period was approximately 80 patients per coach when the program was at full capacity. The coaches were fully integrated into the diabetes clinic care team, ensuring that care plans and individualized goals were achieved. If the coach determined that a care plan was not effectively meeting patient goals, they communicated with the patient's endocrinologist or nurse practitioner to modify the care plan.

### Treatment groups

**Intervention group (DCC patients).** The intervention group consisted of adult patients with diabetes mellitus (type 1 or type 2) who met all of the following criteria: (1) persistent severe hyperglycemia, defined as A1c ≥ 9% on the two most recent results taken at least 3 months apart with the most recent test performed a minimum of three months after the patient's initial visit to the diabetes clinic; (2) ongoing care by an endocrinologist at the diabetes clinic; (3) at least one diabetes education session within the preceding eighteen months or a refusal to attend diabetes education before program enrollment; and (4) enrollment in the DCC program between 2021–2023. A total of 239 patients enrolled in the DCC program during the study period were included in analyses of clinical outcomes. Of these, 43 patients were members of ACO, enabling cost analyses with claims data.

**Comparison group (matched patients).** To serve as a comparison group, we identified diabetes clinic patients who were eligible for the DCC program but not enrolled. These patients had a diagnosis of diabetes (ICD-10 codes E11., E10., or related), persistent poor glycemic control (two A1c values >9.0% at least 3 months apart in the year prior to the index date with no A1c < 9% in that period), and received specialty care at the diabetes clinic (including endocrinologist visits and diabetes education) but never participated in DCC. An index date for each comparison patient was assigned corresponding to the timing of program enrollment for a matched intervention patient to align follow-up periods. We identified 815 eligible comparison patients for the clinical outcomes analysis. For cost analysis, 133 of the comparison group patients were members of ACO with available TME data.

### Propensity score weighting

We used propensity score methods to balance baseline characteristics between the intervention and comparison groups. A propensity score for DCC program enrollment was estimated via logistic regression using patient demographics (age, sex, race, ethnicity), baseline clinical status (Charlson Comorbidity Index and baseline A1c), and prior trends in outcomes (change in A1c over the 1-year pre-enrollment period for all patients; baseline TME, and 1-year pre-enrollment TME trend for ACO patients). Using these propensity scores, we applied optimal full matching with weights (*MatchIt* package in R) to construct a weighted comparison cohort. Each DCC patient was matched to one or more comparison patients with similar

propensity scores, and weights were assigned such that the weighted comparison group resembled the intervention group on the matching variables. This approach builds on FDA's guidance for real-world evidence and has been previously used by our team for similar evaluations of multicomponent interventions [13,14]. Balance on key covariates before and after matching was assessed.

## Outcomes and measures

The primary outcome was change in glycemic control as measured by A1c. For each participant, we extracted all A1c values in the 12-month period prior to DCC enrollment (or index date for controls) and all A1c values in the 12-month post-enrollment period. The difference in means (post-year mean minus pre-year mean) in A1c was calculated for each patient, and then averaged within each treatment group to evaluate mean A1c change between groups. We also examined time trends of A1c over the study period for each group, illustrated in Fig 1. Secondary outcomes included health care utilization in the year following enrollment, including: (a) number of emergency department (ED) visits, (b) number of inpatient hospital admissions, (c) total length of stay (LOS) for hospitalizations, (d) all-cause 30-day readmission (binary, any readmission within 30 days of a hospital discharge during follow-up), and (e) all-cause 90-day readmission. Pre- year and post-year utilization metrics were averaged for use in difference-in-differences analyses. Additionally, for ACO patients with claims data, we examined TME, defined as the sum of all allowed amounts for health care services in a year (inclusive of inpatient, outpatient, professional, and ancillary services). TME were annualized and extreme values were winsorized to account for outliers. We analyzed change in annual TME (post-year minus pre-year) for the intervention versus comparison groups. Financial data on the cost of the DCC program and pharmacy revenues were collected from program records to inform interpretation of economic sustainability.

Intervention Group

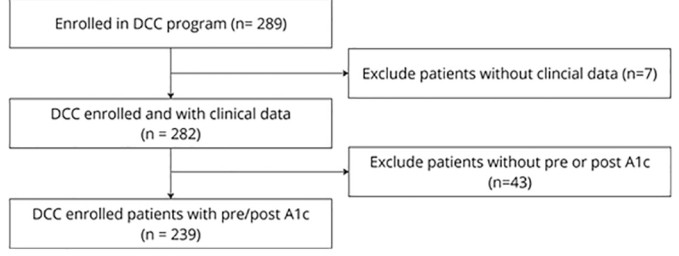

Comparison Group

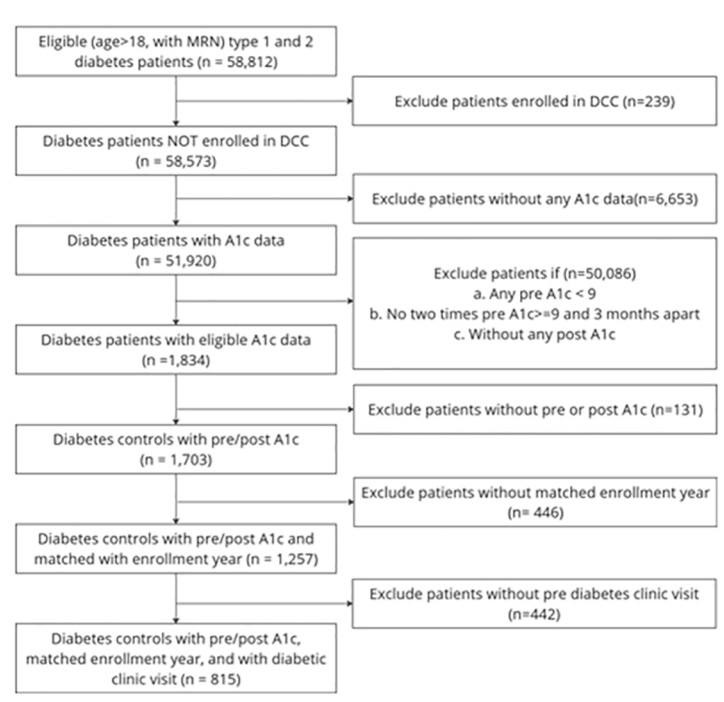

**Fig 1. Consort diagram of Intervention and Comparison groups.**

## Statistical analysis

We used a difference-in-differences (DiD) analytic approach to estimate the impact of the DCC program on outcomes, comparing the change in each outcome from post-year to pre-year between the intervention and comparison groups. For each outcome, we first calculated unadjusted differences in change (Intervention vs Comparison (unweighted)) and conducted hypothesis testing using two-sample t-tests or χ² tests as appropriate. We then fit linear regression models to estimate adjusted differences in outcomes between Intervention and Comparison Group (weighted), accounting for age, sex, race, ethnicity, baseline Charlson Comorbidity Index, and baseline values of the outcome. These models yielded adjusted mean differences in the change from baseline associated with the intervention (i.e., the DiD estimator), along with 95% confidence intervals and p values. Statistical significance was set at a two-sided $\alpha = 0.05$. Given multiple outcomes, we focused on the magnitude and consistency of effects rather than formal multiplicity adjustments, as this was an exploratory evaluation of program impact. All analyses were conducted using R version 4.3.3. We conducted a predefined subgroup analysis stratifying patients by ACO enrollment (those with TME data vs. those without) to explore whether the intervention's effect differed in the context of an ACO care model. The stratified analysis allowed us to assess if the lack of observed cost data for non-ACO patients might bias overall results, by comparing outcome trends in the two strata.

## Results

### Participant characteristics

A total of 239 patients enrolled in the DCC program met inclusion criteria and were matched to 815 comparison patients. Table 1 presents baseline characteristics of the two groups. Before weighting, the Intervention group had a slightly higher comorbidity index and more outpatient visits, but after weighting these differences were minimized (standardized mean differences <0.1 for all covariates). The final weighted sample had a mean age of 57 years, with about half (52% of Intervention and 49% of Comparison) being female, and 61% White and 73% non-Hispanic in each group. All patients had long-standing poor glycemic control, with mean baseline A1c of 10.3% (SD 1.7) in the Intervention group and 10.4% (SD 1.6) in the Comparison group. Baseline rates of ED visits and hospitalizations in the prior year were similarly high in both groups (approximately 0.6 ED visits and 0.6 inpatient admissions per patient on average). Among the 43 Intervention patients in the ACO subset, baseline characteristics were similar to the 133 Comparison participants (Table 2). Baseline

**Table 1. Baseline characteristics of DCC intervention and comparison groups.**

| Variable | Comparison (n = 815) | Intervention (n = 239) |
|---|---|---|
| Age, mean ± SD | 56.7 ± 14.2 | 56.8 ± 14.9 |
| Sex = Women, n (%) | 402 (49.3%) | 125 (52.3%) |
| Race = White, n (%) | 510 (62.6%) | 145 (60.7%) |
| Ethnicity = Non-Hispanic, n (%) | 587 (72.0%) | 175 (73.2%) |
| Baseline A1c Value, mean ± SD | 10.4 ± 1.6 | 10.3 ± 1.7 |

**Table 2. Baseline characteristics of DCC intervention and comparison group participants with TME data (ACO enrollees).**

| Variable | Comparison (n = 133) | Intervention (n = 43) |
|---|---|---|
| Age, mean ± SD | 65.8 ± 12.2 | 64.2 ± 13.6 |
| Sex = Women, n (%) | 65 (48.9%) | 16 (37.2%) |
| Race = White, n (%) | 97 (72.9%) | 31 (72.1%) |
| Ethnicity = Non-Hispanic, n (%) | 96 (72.2%) | 38 (88.4%) |
| Baseline A1c Value, mean ± SD | 10.2 ± 1.4 | 10.2 ± 1.8 |

annual TME in this subset was about $22,100 for the Intervention group and $17,400 for Comparison group. This difference was not statistically significant after weighting.

## Glycemic control

Both the Intervention and Comparison groups showed improvement in glycemic control over the 12-month follow-up, reflecting the impact of intensified diabetes care in this poorly controlled population. The mean A1c decline in Intervention group was –1.5 compared to –1.4 in the Comparison group (Table 3). After weighting, the Intervention group had a 0.4% point larger improvement in A1c than the Comparison group, although this did not reach statistical significance (adjusted DiD –0.4; 95% CI, –1.1 to 0.3; $p=0.20$). Fig 2 illustrates the average A1c trends over time. Both groups' mean A1c levels fell over the first 6 months and plateaued by 12–15 months, with the Intervention group consistently tracking lower than the Comparison group throughout follow-up. The difference in A1c was most pronounced among patients not enrolled in ACO (who lacked additional care coordination support). That subgroup had an adjusted A1c improvement of 0.5% points greater than Comparison group, whereas the ACO-enrolled subgroup saw a 0.2% point greater improvement).

## Health care utilization

The mean number of inpatient admissions in the post-year decreased by 0.03 (SD=1.6) from the pre-year in the Intervention group compared to an increase of 0.07 (SD=1.3) in Comparison group (adjusted difference of –0.2 admissions per patient; 95% CI, –0.4 to 0.04; $p=0.09$). ED visits were unchanged in the Intervention group but increased by 0.08 (SD=1.4) in Comparison group, corresponding to an adjusted difference of –0.2 ED visits per patient (95% CI, –0.4 to 0.1; $p=0.20$). Length of hospital stay showed a notable difference with total days hospitalized increased by 0.5 days in the Intervention group compared to an increase of 2.1 days in Comparison group, an adjusted difference of 3.7 days per patient (95% CI, −8.5 to 1.0). Among non-ACO patients, the adjusted difference in days hospitalized was –5.2 days (95% CI –9.8 to –0.6; $p=0.03$), indicating shorter hospital stays for Intervention group. Differences in rates of hospital readmissions at 30 or 90 days were not statistically significant. While none of the utilization outcomes reached $p<0.05$ in the full cohort, the Intervention group consistently trended toward fewer ED visits and hospitalizations. For every 100 patients in the Intervention group, there were approximately 15 fewer ED visits, 22 fewer hospital admissions, and 3–5 fewer total hospital days over one year compared to 100 Comparison group patients (based on the weighted DiD estimates), though with wide confidence intervals.

**Table 3. Summary of pre, post, and changes in glycemic control, health care utilization, and TME.**

| | Comparison | | | Intervention | | | Difference in Change |
|---|---|---|---|---|---|---|---|
| | Pre | Post | Change | Pre | Post | Change | Adjusted DiD** |
| A1c, mean±SD | 10.2±1.4 | 9.7±2.4 | −1.4±3.7 | 10.2±1.8 | 8.8±1.5 | −1.5±1.7 | −0.4 (−1.1 to 0.3) |
| ER visits, mean±SD | 0.58±1.2 | 0.65±1.3 | 0.08±1.4 | 0.64±1.6 | 0.65±1.3 | 0.00±1.3 | −0.2 (−0.4 to 0.1) |
| Inpatient visits, mean±SD | 0.54±1.3 | 0.65±1.3 | 0.07±1.3 | 0.72±1.5 | 0.65±1.3 | −0.03±1.6 | −0.2 (−0.5 to 0.04) |
| Length of stay, mean±SD | 8.3±15.7 | 10.3±18.7 | 2.1±18.3 | 7.8±9.7 | 8.3±11.8 | 0.5±11.8 | −3.7 (−8.5 to 1.0) |
| All-cause 30-day readmits, mean±SD | 0.36±1.5 | 0.46±1.5 | 0.10±1.5 | 0.48±1.8 | 0.48±1.4 | 0.00±1.6 | −0.2 (−0.4 to 0.1) |
| All-cause 90-day readmits, mean±SD | 0.57±1.8 | 0.73±2.0 | 0.16±1.9 | 0.72±2.4 | 0.76±2.1 | 0.03±2.0 | −0.3 (−0.6 to 0.02) |
| TME*, mean±SD | $14,494±22,849 | $20,057±38,984 | $5,564±29,783 | $19,452±25,322 | $20,272±45,514 | $819±35,876 | $-2,649 ($-14,425 to $9,127) |

*TME data only collected and analyzed for sub-group of participants enrolled in ACO.

**Adjusted DiD models accounting for age, sex, race, ethnicity, baseline Charlson Comorbidity Index, and baseline values of the outcome.

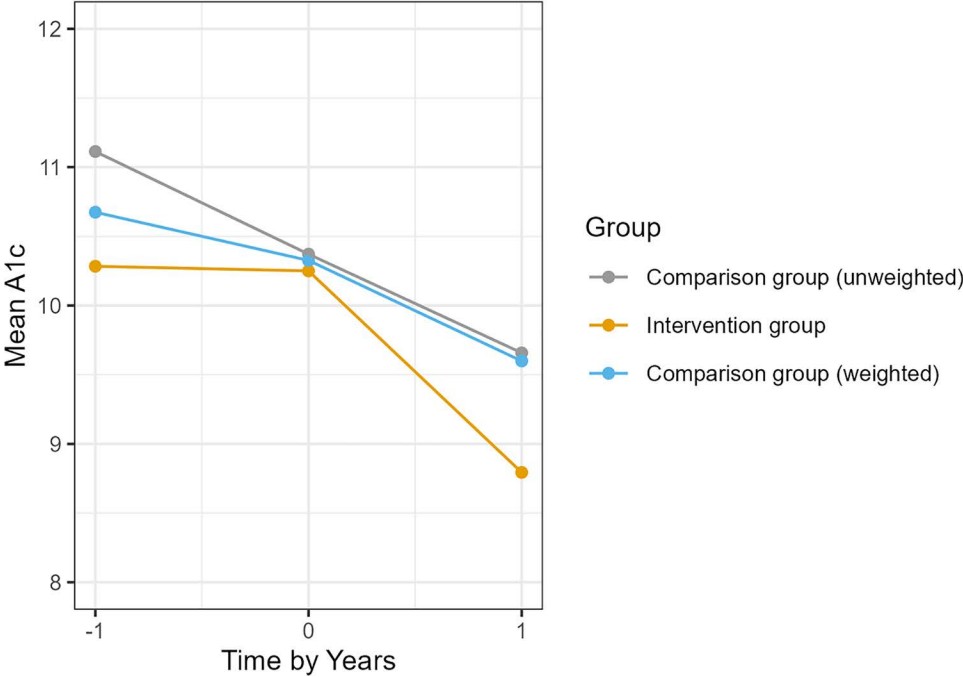

**Fig 2. Change in mean A1c for each group.**

## Total medical expenditure

Baseline mean TME among all ACO patients was high (approximately $22,000 per patient-year), reflecting the complex health needs of this population. Following the intervention, the Intervention group's average annual TME increase by about $819 (from $19,452 to $20,272), whereas the Comparison group's TME increased by about $5,564 (from $14,494 to $20,057). Fig 3 illustrates the average TME trends over time. The adjusted difference-in-differences was a reduction of $2,649 per patient per year in the Intervention group compared to Comparison group (95% CI, –$14,425 to $9,127). This corresponds to a 12% relative reduction in TME associated with the DCC program, although this estimate was not statistically significant ($p = 0.66$) due to high variability and the small sample. Unadjusted analyses similarly suggested possible cost savings (unadjusted mean difference between groups of $4,745 per patient, 95% CI –$15,851 to $6,363; $p = 0.40$), but with wide variability. These exploratory cost results imply a trend toward lower health care spending for Intervention group patients, aligning with the observed reductions in hospital utilization. Confirmatory analyses with a larger sample would be needed to reach statistical significance. It is notable that the DCC program's annual per-patient cost was approximately $2,800, which is of similar magnitude to the observed $2,649 average savings in medical expenditure. Although not part of the formal outcomes analysis, internal financial records indicated that DCC patients' improved medication adherence and engagement generated additional revenue through the affiliated specialty pharmacy (including 340B drug pricing savings). This pharmacy-derived revenue was sufficient to offset the program costs on a per-patient basis. In other words, better adherence and more prescriptions filled within the health system produced net income that covered the DCC staffing expenses, an important consideration for sustainability.

## ACO subgroup analyses

Results stratified by ACO enrollment revealed that the intervention effects were consistently in the favorable direction for both subgroups, but with different magnitudes. Among patients enrolled in the ACO, the Intervention group showed a

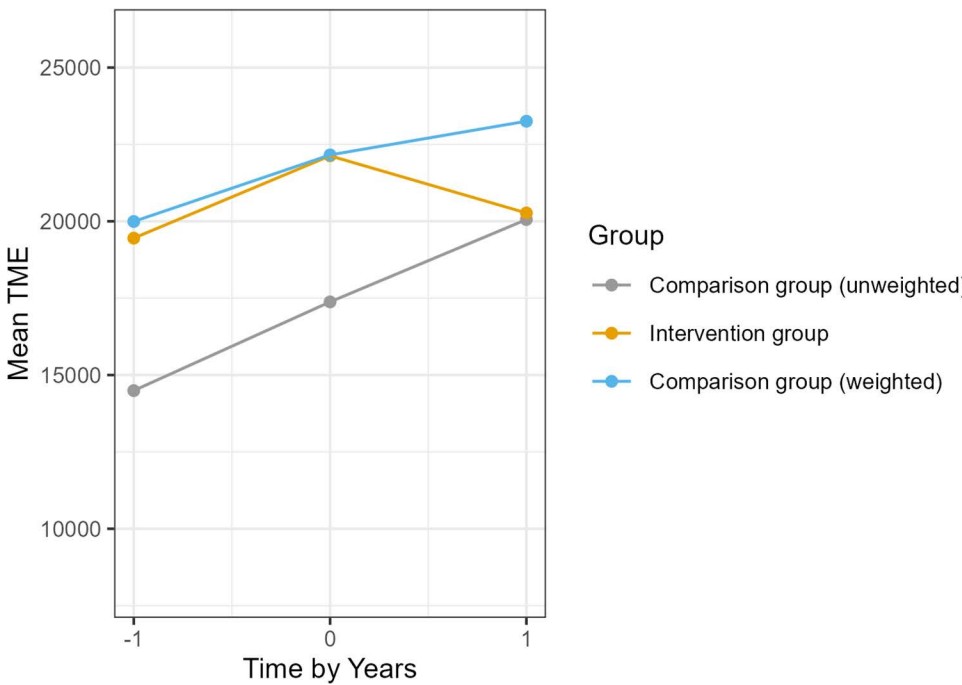

**Fig 3. Change in TME for each group.**

smaller relative improvement. For example, the adjusted A1c reduction was –0.16 (95% CI –0.90 to 0.59) in this group, and there were slight, nonsignificant increases in utilization compared to Comparison group patients (e.g., a 0.37 increase in admissions, 95% CI –0.38 to 1.1). In contrast, among patients not in the ACO (fee-for-service group without ACO care coordination), the DCC program's impact was more pronounced: adjusted mean A1c improvement of –0.50% (–1.3 to 0.27), and notable decreases in utilization, such as –0.26 admissions (95% CI –0.54 to 0.01, $p = 0.06$) and –0.21 ED visits (95% CI –0.45 to 0.03, $p = 0.08$) per patient. Notably, in the non-ACO subgroup (n = 196), DCC enrollment was associated with significantly shorter hospitalization LOS (–5.18 days, 95% CI –9.8 to –0.58; $p = 0.03$) and lower 90-day readmission rates (unadjusted weighted difference –0.41 per patient, 95% CI −0.77 to –0.04; $p = 0.03$). These findings suggest that the benefits of the DCC program were even stronger in the general diabetes clinic population than in the ACO-managed population. Importantly, this supports the generalizability of the program's positive trends to patients for whom we did not have claims data for, reinforcing confidence that meaningful cost savings likely extend to the broader group of high-risk patients beyond the ACO subset.

## Discussion

In this retrospective cohort study of adults with poorly controlled diabetes, we found that a pharmacist driven telehealth coaching program integrated within a specialty clinic was associated with trends towards improved clinical outcomes and potential reductions in health care utilization compared to usual care. Over one year, coached patients achieved a greater reduction in hemoglobin A1c (an additional 0.5% drop on average) than matched comparisons, although this difference was not statistically significant. The coaching intervention group also consistently trended toward fewer ED visits and hospital admissions. And in subgroup analysis of patients outside the ACO care model, the DCC program was associated with significantly shorter hospital stays and lower readmission rates. While the observed differences in utilization did not reach statistical significance, their magnitude (15 fewer ED visits and 22 fewer hospitalizations per 100 patients) suggests

clinically meaningful impacts. Furthermore, we observed a reduction in TME averaging roughly $2,650 (12%) per patient per year in the DCC group relative to comparison patients. Notably, this magnitude of cost savings was similar to the program's operational cost. Additionally, there was enhanced pharmacy revenue from patients choosing to fill at specialty pharmacy and improved medication adherence in the DCC program. Taken together, these findings indicate that the DCC program supported improved glycemic control and may have contributed to reductions in acute care utilization in a high-risk population, doing so in a manner that was financially sustainable for the health system. Even without definitive statistical significance for all outcomes, the consistency of positive trends across clinical, utilization, and cost metrics supports the potential benefit. Our two-year A1c data for DCC patients shows a similar reduction from baseline to one-year. Longer follow-up will be important to determine whether these trends are sustained over time.

Our results align with and extend prior evidence on diabetes coaching and pharmacist-led interventions. The magnitude of A1c reduction observed in DCC participants (approximately 1.5 percentage points from baseline) is substantial in absolute terms and comparable to improvements reported in randomized trials of intensive coaching. However, because usual care patients in our study also showed ~1.5% A1c improvement (likely due to concurrent specialist care and regression to the mean), the comparative effect of the DCC program was reduced (0.4%). This pattern is consistent with some real-world studies where both intervention and comparison groups improve over time, reducing detectable between-group differences. In structured clinical trials, coaching interventions have achieved net A1c reductions in the range of 0.3–0.8% beyond usual care. A mobile health coaching trial by Gerber et al. showed a 0.8% greater A1c reduction in the intervention arm [6]. Our findings of a 0.4% relative A1c improvement fall within this expected range, considering that our comparison group was not "untreated" but receiving specialty endocrinology care. The public health relevance is clear: each 0.5-point population level shift toward glycemic control among this group of patients with intransigent A1c values helps the United States move closer to the Healthy People 2030 objective of reducing the proportion of adults with A1c > 9%.

Our subgroup analysis illuminates a plausible pathway for these benefits. Among patients not enrolled in a Medicare ACO, and therefore lacking baseline care-coordination infrastructure, the program produced larger clinical gains and statistically significant reductions in hospital days and 90-day readmissions. By contrast, incremental effects were smaller in ACO enrollees who already received case-management services. These observations imply that the DCC's high-frequency telehealth contacts, frequent medication titration, and rapid problem-solving filled a critical coordination gap for fee-for-service patients, thereby attenuating crises that would otherwise escalate to emergency use. The finding aligns with the broader literature linking diabetes coaching to improved self-management and extends it by demonstrating real-world impact when coaching augments, not duplicates, existing population health efforts.

The program's mean reduction in TME, although non-statistically significant, of $2,649 (12%) per patient in annual TME among ACO participants approximated its operating cost of about $2,800 per patient. Importantly, internal financial data showed that improved medication adherence yielded additional specialty-pharmacy and 340B revenue sufficient to offset program costs. This break-even profile suggests that pharmacist-led coaching may be capable of self-financing under current payment models. As value-based contracts mature, any future shared-savings dollars or revenue retained from avoided utilization could be reinvested in services that address the social drivers of uncontrolled diabetes, such as transportation vouchers, healthy-food prescriptions, or broadband access for telehealth.

Interpretation should account for several limitations. First, despite rigorous propensity weighting and DiD analyses, residual confounding is possible in this non-randomized study. Referral to the DCC program was provider-initiated and patient enrollment was voluntary, introducing potential selection bias that may not be fully addressed by propensity matching. It is possible that care disruption caused by the COVID-19 pandemic led to an increase in number of patients with uncontrolled diabetes and decentralized nature of DCC provided a viable source of care. However, these extrinsic factors were unlikely to differentially bias participation. Second, claims data were available only for ACO patients, constraining statistical power for cost outcomes. Nevertheless, the directionally larger clinical and healthcare utilization effects in non-ACO patients bolster confidence that cost savings may be at least comparable in fee-for-service populations.

Third, the study was conducted at a single health system; replication elsewhere will clarify generalizability. Finally, we only assessed one-year outcomes. Longer follow-up is needed to determine whether observed trends translate into sustained clinical improvements and cost reductions.

Taken together, these findings suggest that multidisciplinary diabetes coaching can meaningfully advance national goals to lower the burden of very high A1c, even when delivered within the constraints of everyday clinical operations. The absence of conventional statistical significance for several utilization metrics does not negate their clinical salience, particularly in the context of a pragmatic real-world evaluation. Preventing even a handful of admissions in a high-risk cohort generates patient benefit and offsets coaching costs. Future research should test the model prospectively, quantify patient-reported outcomes, and evaluate explicit reinvestment of 340B or shared-savings proceeds into social needs interventions. Meanwhile, health systems without robust care-coordination programs may realize the greatest return by targeting pharmacist-coach resources to patients whose uncontrolled diabetes currently drives avoidable utilization and expense.

## Author contributions

**Conceptualization:** Daniel Amante, Samir Malkani, Richard Haas, Cheryl Barry, Bill McElnea, Lillian Piz, Gabriella Pugliese, Apurv Soni.

**Formal analysis:** Biqi Wang.

**Writing – original draft:** Daniel Amante, Samir Malkani, Richard Haas, Biqi Wang, Cheryl Barry, Bill McElnea, Lillian Piz, Gabriella Pugliese, Hinal Sharma, Apurv Soni.

**Writing – review & editing:** Daniel Amante, Samir Malkani, Richard Haas, Biqi Wang, Cheryl Barry, Bill McElnea, Lillian Piz, Gabriella Pugliese, Hinal Sharma, Apurv Soni.

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
