## [Decision Letter · Decision Letter 0]

11 Dec 2025

PONE-D-25-56341

Effectiveness of a pharmacist diabetes coaching program: a propensity-matched retrospective analysis.

PLOS One

Dear Dr. Amante,

Thank you for submitting your manuscript to PLOS ONE. After careful consideration, we feel that it has merit but does not fully meet PLOS ONE’s publication criteria as it currently stands. Therefore, we invite you to submit a revised version of the manuscript that addresses the points raised during the review process.

We look forward to receiving your revised manuscript.

Kind regards,

Yee Gary Ang, MBBS MPH

Academic Editor

PLOS One

**Journal Requirements:**

“The Diabetes Care Coach program is funded in part by Shields Health Solutions, a vendor to UMass Memorial Health. Shields-affiliated authors on this study have no other financial associations with UMass Memorial Health.

This project was partially supported by the National Institute of Diabetes and Digestive and Kidney Diseases (NIDDK) Grant K01DK131318 (support for DA).

AS has consulted for ARPA-H, Pfizer, FDA, and Shields Health Solutions on matters unrelated to this manuscript.

This project was also partially supported by the National Center for Advancing Translational Sciences (NCATS), National Institutes of Health, through Grant UL1TR001453 (support for AS). The content is solely the responsibility of the authors and does not necessarily represent the official views of the NIH.”

4. In the online submission form, you indicated that your data is available only on request from a third party. Please note that your Data Availability Statement is currently missing contact details for the third party, such as an email address or a link to where data requests can be made. Please update your statement with the missing information.

5. We note that you have indicated that there are restrictions to data sharing for this study. PLOS only allows data to be available upon request if there are legal or ethical restrictions on sharing data publicly. For more information on unacceptable data access restrictions, please see http://journals.plos.org/plosone/s/data-availability#loc-unacceptable-data-access-restrictions.

6. Thank you for stating the following in the Financial Disclosure section:

“The Diabetes Care Coach program is funded in part by Shields Health Solutions, a vendor to UMass Memorial Health. Shields-affiliated authors on this study have no other financial associations with UMass Memorial Health. This project was partially supported by the National Institute of Diabetes and Digestive and Kidney Diseases (NIDDK) Grant K01DK131318 (support for DA). AS has consulted for ARPA-H, Pfizer, FDA, and Shields Health Solutions on matters unrelated to this manuscript. This project was also partially supported by the National Center for Advancing Translational Sciences (NCATS), National Institutes of Health, through Grant UL1TR001453 (support for AS). The content is solely the responsibility of the authors and does not necessarily represent the official views of the NIH.”

We note that one or more of the authors are employed by a commercial company: Shields Health Solutions.

A. Please provide an amended Funding Statement declaring this commercial affiliation, as well as a statement regarding the Role of Funders in your study. If the funding organization did not play a role in the study design, data collection and analysis, decision to publish, or preparation of the manuscript and only provided financial support in the form of authors' salaries and/or research materials, please review your statements relating to the author contributions, and ensure you have specifically and accurately indicated the role(s) that these authors had in your study. You can update author roles in the Author Contributions section of the online submission form.

B. Please also provide an updated Competing Interests Statement declaring this commercial affiliation along with any other relevant declarations relating to employment, consultancy, patents, products in development, or marketed products, etc.

7. Please upload a new copy of Figure 1 as the detail is not clear. Please follow the link for more information: https://journals.plos.org/plosone/s/figures

8. Please ensure that you refer to Figures 2 and 3 in your text as, if accepted, production will need this reference to link the reader to the figure.

**Additional Editor Comments:**

Thank you once again for submitting your research to PLOS One. It was a pleasure to read your work.

We invited multiple reviewers to assess your manuscript. Two reviewers have returned their evaluations, which present mixed feedback: one has raised major concerns whilst the other has provided minor comments.

Please review the attached comments carefully and assess whether you are able to address the reviewers' points. If feasible, we would welcome a resubmission of your manuscript incorporating the necessary revisions.

Reviewers' comments:

Reviewer's Responses to Questions

**Comments to the Author**

1. Is the manuscript technically sound, and do the data support the conclusions?

Reviewer #1: Yes

Reviewer #2: Yes

2. Has the statistical analysis been performed appropriately and rigorously?

Reviewer #1: Yes

Reviewer #2: Yes

3. Have the authors made all data underlying the findings in their manuscript fully available?

Reviewer #1: Yes

Reviewer #2: Yes

4. Is the manuscript presented in an intelligible fashion and written in standard English?

Reviewer #1: Yes

Reviewer #2: Yes

5. Review Comments to the Author

Reviewer #1: Dear authors,

Thank you for the opportunity to review this very interesting manuscript. I found it clearly written, well-organized, and timely study that addresses an important gap in real-world evidence regarding pharmacist-led telehealth coaching models for patients with uncontrolled diabetes (A1c ≥ 9%). The study rationale is strong, the methods are appropriate, and the findings demonstrate encouraging trends toward improved glycemic outcomes and reduced acute care utilization. The manuscript makes a valuable contribution to the literature by evaluating program performance within an operational health system context rather than a controlled trial environment.

The analytical approach, including propensity-score matching and difference-in-differences analysis, is clearly presented and appropriate for the research question. The exploration of financial implications, particularly the use of 340B revenue mechanisms, provides meaningful insight into sustainability considerations for health system decision-makers. The conclusions are generally reasonable and aligned with the data presented.

However, several important limitations require stronger acknowledgment and clarification. Because participation in the program was voluntary rather than randomized, there is a significant risk of selection bias; participants who agreed to enroll may differ systematically from nonparticipants in motivation, engagement, health literacy, or social support. Although matching methods reduce measured imbalance, they cannot address unmeasured differences. A more explicit discussion of this limitation would strengthen transparency and interpretability. Additionally, many core outcomes—including A1c reduction and cost estimates—did not reach statistical significance, and the economic analysis is based on only 43 ACO patients, limiting precision. Cost-effectiveness claims should therefore be interpreted with caution.

In addition, important details regarding the intervention itself are insufficiently described. The manuscript does not specify how many pharmacists served as coaches, their full-time equivalency, or the patient-to-coach staffing ratio. This information is essential for assessing scalability and resource requirements. Without clarity on workforce size, it is difficult for readers to gauge feasibility or attempt replication in other settings. Similarly, although pharmacists are described as having “additional training” in diabetes management, the manuscript does not provide information about the content, duration, credentialing, or cost of this training, and training costs were not included in the financial analysis. More detail would improve understanding of implementation complexity and startup costs. The frequency, duration, and structure of telehealth encounters are also not reported, limiting reproducibility.

The manuscript would also benefit from reporting the total number of diabetes patients served by the UMMH system or at the Diabetes Center to contextualize the sample relative to the broader population. Finally, because workforce configuration is central to program design, a brief justification for selecting pharmacists rather than nurses, physicians, or mixed models would improve clarity.

Despite these limitations, this is an important and promising program evaluation, and the manuscript provides useful early evidence supporting multidisciplinary telehealth approaches for high-risk chronic disease management. With clearer description of intervention structure, staffing, and limitations, the manuscript would offer even stronger guidance to health systems seeking to implement or scale similar programs.

Thank you again for the opportunity to review this thoughtful contribution.

Reviewer #2: This is a well written paper in a topic area that needs further evidence in the literature base. Obtaining this type of data is difficult, and I am glad this study was conducted. I was also impressed the authors conducted the ACO subgroup analyses that was an insightful layer of information.

The only suggestion I have is to present the results in the figures in table format as well. The dollar values are important to many stakeholders, and being able to compare them numerically will be valuable. I would suggest a table demonstrating changes in glycemic control, health care utilization (by inpatient, ED, LOS, etc) and TME. These results are presented in the text, but a table format would be very insightful.

The timeline also begins in 2020, can the authors clarify how they adjusted for COVID during that year?

6. PLOS authors have the option to publish the peer review history of their article (what does this mean? ). If published, this will include your full peer review and any attached files.

**Do you want your identity to be public for this peer review?**  For information about this choice, including consent withdrawal, please see our Privacy Policy .

Reviewer #1: **Yes:** Akihiro Seita

Reviewer #2: No

---

## [Author Response · Author response to Decision Letter 1]

15 Jan 2026

We appreciate the thoughtful and constructive feedback provided by the Academic Editor and reviewers. We have carefully considered all comments and revised the manuscript accordingly. The uploaded Response to Reviewers document contains a table that presents a point-by-point response to each comment, including the verbatim comment, our response, a description of the revisions made, and the location(s) of these revisions in the manuscript. Changes are also highlighted in the accompanying tracked-changes version of the manuscript.

---

## [Decision Letter · Decision Letter 1]

9 Feb 2026

Dear Dr. Amante,

Thank you for submitting your manuscript to PLOS ONE. After careful consideration, we feel that it has merit but does not fully meet PLOS ONE’s publication criteria as it currently stands. Therefore, we invite you to submit a revised version of the manuscript that addresses the points raised during the review process.

**ACADEMIC EDITOR:**

We have invited multiple reviewers

2 of them have responded and have given some comments

Please resubmit if you are able to address them

We look forward to receiving your revised manuscript.

Kind regards,

Yee Gary Ang, MBBS MPH

Academic Editor

PLOS One

Journal Requirements:

Additional Editor Comments:

We have invited multiple reviewers

2 of them have responded and have given some comments

Please resubmit if you are able to address them

Reviewers' comments:

Reviewer's Responses to Questions

**Comments to the Author**

Reviewer #1: (No Response)

Reviewer #3: All comments have been addressed

2. Is the manuscript technically sound, and do the data support the conclusions?

Reviewer #1: Yes

Reviewer #3: Yes

3. Has the statistical analysis been performed appropriately and rigorously?

Reviewer #1: Yes

Reviewer #3: Yes

4. Have the authors made all data underlying the findings in their manuscript fully available?

Reviewer #1: Yes

Reviewer #3: Yes

5. Is the manuscript presented in an intelligible fashion and written in standard English?

Reviewer #1: Yes

Reviewer #3: Yes

Reviewer #1: Dear Authors,

Thank you for the revised manuscript and for your careful efforts to address the reviewer comments. The study remains timely, clearly written, and methodologically sound, and it provides valuable real-world insight into pharmacist-led telehealth support for patients with poorly controlled diabetes. The strengthened discussion of study limitations, particularly regarding the non-randomized design and limited statistical power for cost analyses, is appreciated.

To further strengthen the manuscript and improve its utility for readers and health system decision-makers, I encourage the authors to consider the following clarifications and additions.

First, the telehealth intervention itself would benefit from more detailed description. Please consider explicitly reporting the number of pharmacist coaches involved in the program, the full-time equivalent staffing dedicated to the intervention, and the resulting patient-to-coach ratio. These details are essential for assessing feasibility, scalability, and resource requirements, and would greatly enhance reproducibility.

Second, additional information on pharmacist training would improve transparency. Specifically, please describe the content, duration, credentialing, and approximate cost of the additional diabetes-related training received by the pharmacists, and clarify whether training costs were included in or excluded from the financial analysis. This would help readers understand implementation complexity and startup costs.

Third, greater clarity regarding telehealth delivery would strengthen interpretation of the findings. Please consider reporting the typical frequency, duration, and structure of telehealth encounters, even if these varied across patients. If the intervention was intentionally flexible rather than standardized, this should be stated explicitly. Where possible, summarizing the distribution of telehealth contacts, such as the median number of encounters per patient, would allow readers to better interpret program dose and its potential relationship to outcomes.

Finally, I encourage careful framing of outcomes. While the reduction in hospital length of stay among non-ACO patients is clinically meaningful and should be highlighted, the primary clinical outcome and most utilization and cost outcomes did not reach statistical significance. Presenting these results as exploratory or directional rather than confirmatory would align conclusions more closely with the data.

With these clarifications, the manuscript would provide clearer and more actionable guidance for health systems considering implementation or scale-up of pharmacist-led telehealth models for high-risk diabetes management.

Thank you again for the opportunity to review this important and promising work.

Reviewer #3: thank you for investing time and effort in understanding the intent behind the suggestions and comments from the preceding round of review.

i note that due effort has been made to elaborate on the methodological description with a view to greater transparency of reporting; likewise the limitations and the funding statements have been improved in the same spirit.

the improvements made greatly address an earlier critique of the manuscript, namely that too much was assumed of the eventual readership. in my opinion, the present iteration of the manuscript addresses this critique adequately.

**Do you want your identity to be public for this peer review?** For information about this choice, including consent withdrawal, please see our Privacy Policy

Reviewer #1: No

Reviewer #3: **Yes:** Kenneth Y T Lim

---

## [Author Response · Author response to Decision Letter 2]

25 Feb 2026

We appreciate the thoughtful and constructive feedback provided by the Academic Editor and reviewers. We have carefully considered all comments and revised the manuscript to improve clarity, transparency, and implementation relevance. Revisions focused on strengthening the description of the Diabetes Care Coach intervention, including staffing structure, pharmacist training context, patient-to-coach ratios, and telehealth delivery (frequency, duration, and flexible structure), as well as clarifying the typical cadence of patient engagement. We also refined the Abstract and Discussion to more appropriately frame findings as directional and consistent with a pragmatic observational evaluation, particularly in light of non-significant outcomes. A detailed, point-by-point response to each reviewer comment, including verbatim comments, responses, revisions made, and locations within the manuscript, is provided in the table below, with all changes highlighted in the accompanying tracked-changes version.

---

## [Decision Letter · Decision Letter 2]

9 Mar 2026

Effectiveness of a pharmacist diabetes coaching program: a propensity-matched retrospective analysis.

PONE-D-25-56341R2

Dear Dr. Amante,

We’re pleased to inform you that your manuscript has been judged scientifically suitable for publication and will be formally accepted for publication once it meets all outstanding technical requirements.

Kind regards,

Yee Gary Ang, MBBS MPH

Academic Editor

PLOS One

Additional Editor Comments (optional):

Reviewers' comments:

Reviewer's Responses to Questions

**Comments to the Author**

Reviewer #1: All comments have been addressed

2. Is the manuscript technically sound, and do the data support the conclusions?

Reviewer #1: Yes

3. Has the statistical analysis been performed appropriately and rigorously?

Reviewer #1: Yes

4. Have the authors made all data underlying the findings in their manuscript fully available?

Reviewer #1: Yes

5. Is the manuscript presented in an intelligible fashion and written in standard English?

Reviewer #1: Yes

Reviewer #1: Dear Authors,

Thank you for the revised manuscript and for your careful and constructive responses to the reviewer comments. The revisions have substantially improved the clarity and transparency of the manuscript.

In my previous review, I encouraged you to consider providing additional details regarding the telehealth intervention, including the number of pharmacist coaches involved, staffing levels and patient-to-coach ratio, the structure and delivery of telehealth encounters, and further clarification on pharmacist training and the treatment of training costs in the financial analysis. I also suggested careful framing of the study outcomes given that several utilization and cost outcomes did not reach statistical significance.

The revised manuscript addresses these points well. You have added useful information on staffing and the telehealth delivery model, expanded the description of pharmacist training, clarified the treatment of training costs, and appropriately refined the framing of the results to better reflect the exploratory nature of the findings.

Overall, the manuscript now provides a clearer and more informative description of the intervention and its implementation, and it offers valuable insights for health systems considering pharmacist-led telehealth support for high-risk diabetes management.

**Do you want your identity to be public for this peer review?** For information about this choice, including consent withdrawal, please see our Privacy Policy

Reviewer #1: **Yes:** Akihiro Seita

---

## [Editor Report · Acceptance letter]

PONE-D-25-56341R2

PLOS One

Dear Dr. Amante,

I'm pleased to inform you that your manuscript has been deemed suitable for publication in PLOS One. Congratulations! Your manuscript is now being handed over to our production team.

Kind regards,

on behalf of

Dr. Yee Gary Ang

Academic Editor

PLOS One